# Flexural Characteristics of Functionally Graded Fiber-Reinforced Cementitious Composite with Polyvinyl Alcohol Fiber

**Toshiyuki Kanakubo** [1,*] **, Takumi Koba** [2] **and Kohei Yamada** [3]

1 Division of Engineering Mechanics and Energy, University of Tsukuba, Tsukuba 305-8577, Japan
2 Degree Programs in Systems and Information Engineering, University of Tsukuba, Tsukuba 305-8577, Japan; s2020844@s.tsukuba.ac.jp
3 Department of Engineering Mechanics and Energy, University of Tsukuba, Tsukuba 305-8577, Japan; s1820947@s.tsukuba.ac.jp
* Correspondence: kanakubo@kz.tsukuba.ac.jp; Tel.: +81-29-853-5045

**Abstract:** The objective of this study is to investigate the flexural characteristics of functionally graded fiber-reinforced cementitious composite (FG-FRCC) concerning the fiber volume fraction ($V_f$) varying in layers and the layered effect in bending specimens. The FG-FRCC specimens, in which $V_f$ increases from 0% in the compression zone to 2% in the tensile zone, are three-layered specimens using polyvinyl alcohol (PVA) FRCC that are fabricated and tested by a four-point bending test. The maximum load of the FG-FRCC specimens exhibits almost twice that of homogeneous specimens, even when the average of the fiber volume fraction in the whole specimen is 1%. The result of the section analysis, in which the stress–strain models based on the bridging law (tensile stress–crack width relationship owned by the fibers) consider the fiber orientation effect, shows a good adaptability with the experiment result.

**Keywords:** fiber-reinforced cementitious composite; functionally graded material; bending test; fiber volume fraction; section analysis; bridging law



## 1. Introduction

Conventional cementitious composites such as concrete and mortar generally show brittle behavior with cracks opening under the tensile field. Fiber-reinforced cementitious composites (FRCCs)—in which short discrete fibers are mixed into the cementitious matrix with several percentages of fiber volume fraction—have been focused and utilized being expected to present high fracture toughness, deformability and small crack openings. The features of FRCCs can bring high durability for environmental attacks and additional performance for the structural behavior of concrete structures. A number of studies have been conducted to investigate and evaluate the mechanical characteristics of FRCCs, especially for those under flexural and tensile fields.

In the early 1960s, the use of steel fiber in concrete mixture started resulting in improvement of the ductility and fracture toughness of the concrete [1]. On the other hand, large-scale use of polymeric fibers started from the later 1970s [2]. Polymeric fibers have the advantage of non-corrosion characteristics and have been used to control cracking in the early stages of concrete setting. Furthermore, since the 1980s, studies on high-performance fiber-reinforced cement composite (HPFRCC), strain-hardening cement composite (SHCC) and engineered cementitious composite (ECC) have been conducted along with the development of the high strength polymeric fibers such as polyethylene (PE) and polyvinyl alcohol (PVA) fibers [3,4]. These FRCCs exhibit the pseudo strain-hardening behavior and multiple cracking under tensile fields. These have been utilized in the actual applications such as decks and slabs, tunnel linings, seismic walls, and so on [5–7]. It can be pointed out, however, that the costs of fibers used in FRCCs is relatively expensive comparing with

those of other materials in concrete [8]. For example, the price of polymeric fibers for 1% of volume fraction is twice or more than that of conventional concrete. It has been expected to expand the use of these FRCCs with additional values in the appropriate cost performance.

Functionally graded materials (FGMs) have shown that a continuous gradient of material properties is exhibited by the non-homogenous material structures [9,10]. In the engineering materials fields, FGMs have been developed for the industrial applications such as aerospace, automotive, power generation, and so on [11–14]. As one of the typical examples, laminate composites were developed to reduce the thermal stresses due to the huge differences between outside and inside temperatures of engines [15].

Some studies concerning FGMs can be found out in the fields of cementitious composites including fiber-reinforced concrete (FRC). Roesler et al. [16] investigated the improvement of the properties for concrete pavements by distinct layers within the concrete pavement surface. The combinations of layered plain and polymeric FRC brought the peak loads and initial fracture energy improvements relative to full-depth plain concrete. Naghibdehi et al. [17] studied the flexural performance of functionally graded FRC using polypropylene (PP) and steel fibers. They concluded that the lower cost of PP fibers would suggest the use of double-type (two-layered) FRC with steel and PP fibers. Naghibdehi et al. [18] also studied the flexural behavior of the same functionally graded FRC under cyclic loading. Their results indicate that the use of the functionally graded FRC increases the dissipated energy due to the applied cyclic loading. Shen et al. [19] tested bending beam specimens with four-layered and increasing fiber volume fraction of PVA fiber linearly from 0% in the compression zone to 2% in the tensile zone by the extrusion technique. The functionally graded specimens exhibited about 50% higher strengths compared to homogeneous specimens with the same overall fiber volume fraction.

Recently developed FRCCs with high tensile deformability have a high viscosity to distribute fine polymeric fibers randomly. In addition, a self-consolidating property is commonly adopted to have a good workability in casting. These fresh properties of FRCCs affect the fiber orientation in the matrix [20–22], resulting in mechanical characteristics differences due to the fiber orientation [23–25]. Li and Wang [26] explained the fiber orientation as two-dimensional random and three-dimensional random by the dimensions of the specimen. The ultimate tensile strain of ECC tended to decrease as the specimen dimensions change from two-dimensional random to three-dimensional random. As described before, functionally graded fiber-reinforced cementitious composites (FG-FRCCs) have been studied by layered specimens changing fiber type and/or fiber volume fraction in each layer. Even if the fiber types and volume fractions do not vary in layers, a thin layer would cause an improvement in mechanical characteristics of FG-FRCCs by fiber orientation effect.

The objective of this study is to investigate the flexural characteristics of FG-FRCC from the perspective of the effect by fiber volume fraction varying in each layer and both layered effect in bending specimens. For the former point, the fiber volume fraction of three-layered specimens increases from 0% in the compression zone to 2% in the tensile zone, and the results are compared with those obtained from the homogeneous specimens with 1% volume fraction. For the latter point, three-layered specimens with the same volume fraction of 1% in each layer are tested and the results are also compared with the homogeneous specimens. The target FRCC is that with PVA fiber, having a self-consolidating property and precisely the same materials and mixture proportion as the authors' previous study [27]. All the specimens are fabricated by pouring FRCC into the molds considering the simplest fabrication method to be able to adopt in the practical use. In addition to the comparing of the test results in each test parameter, the section analysis is conducted to evaluate the maximum bending moment of FG-FRCC based on the models of bridging law (tensile stress–crack width relationship owned by fibers). The models of bridging law featuring both the volume fraction of PVA fiber and fiber orientation intensity are utilized.

## 2. Experimental Program

### 2.1. Tested FRCC

The mixture proportion of the cementitious matrix used in this study is shown in Table 1. The water binder ratio was set to 0.39, and fly ash and fine sand under 0.2 mm size were utilized. Even mixed with fine fiber of 0.1 mm diameter, it keeps self-consolidating property until around 3% volume fraction of fiber. Table 2 lists the dimensions and mechanical properties of PVA fiber used in this study. These cementitious matrix and fiber are same as those used in the previous study [27]. The compressive strength of the 100 mm diameter–200 mm height cylinder test pieces ranged from 41.1 MPa to 46.4 MPa at the duration of the bending test.

**Table 1.** Mixture proportion.

| Water by Binder Ratio | Sand by Binder Ratio | Unit Weight (kg/m$^3$) | | | |
|:---:|:---:|:---:|:---:|:---:|:---:|
| | | Water | Cement | Fly Ash | Sand |
| 0.39 | 0.50 | 380 | 678 | 291 | 484 |
| Cement: High early strength Portland cement Fly ash: Type II of Japanese Industrial Standard (JIS A 6201) [28] Sand: Size under 0.2 mm High-range water-reducing admixture: Binder $\times$ 0.6% | | | | | |

**Table 2.** Dimensions and mechanical properties of polyvinyl alcohol (PVA) fiber.

| Type | Density (g/cm$^3$) | Diameter (mm) | Length (mm) | Tensile Strength (MPa) | Elastic Modulus (GPa) |
|:---:|:---:|:---:|:---:|:---:|:---:|
| PVA | 1.30 | 0.10 | 12 | 1200 | 28 |

Tested values by manufacturer.

### 2.2. Specimens

The dimensions of all specimens are $100 \times 100$ mm$^2$ cross-section and 400 mm length, to be subjected to four-point bending test according to ISO 21914 [29]. The four-point bending test, in which the ratio of depth to span of the specimen is 1 by 3, is the most commonly-used method to determine flexural characteristics of cementitious composites including concrete [30]. The main test parameter is fiber volume fraction, $V_f$, in three-layered specimens in which the height of each layer is equal to one-third the whole height of the cross-section. The list of the specimens is shown in Table 3 and illustrated in Figure 1. The specimens were grouped into two series by the casting days of FRCC, i.e., 1st day and 2nd day. The specimens in the 1st day were fabricated to compare the effect of fiber volume fraction in FG-FRCC, namely FG-FRCC specimens were layered with 0% (mortar), 1%, and 2% volume fraction from the compression zone to the tensile zone, respectively. The homogeneous specimens with 1% (Hmg-1%) and 2% (Hmg-2%) volume fraction were also fabricated. The specimens in the 2nd day were fabricated to compare the effect of the layer. Layer-1% and Layer-2% specimens were only layered with the same volume fraction of 1% and 2%, respectively. The homogeneous specimens were also fabricated. The FRCC with each fiber volume fraction in the same day casting was poured into the mold from the same mixing batch. The number of the specimens with each test parameter is three.

**Table 3.** List of specimens.

| Test Series (Casting Day) | Specimen ID | Remarks | Fiber Volume Fraction, $V_f$ | Number of Specimens |
|---|---|---|---|---|
| 1st day | FG-FRCC | Functionally graded | 0, 1, 2% | 3 for each parameter |
| | Hmg-1% | Homogeneous | 1% | |
| | Hmg-2% | Homogeneous | 2% | |
| 2nd day | Layer-1% | Three-layer | 1% | 3 for each parameter |
| | Layer-2% | Three-layer | 2% | |
| | Hmg-1% | Homogeneous | 1% | |
| | Hmg-2% | Homogeneous | 2% | |

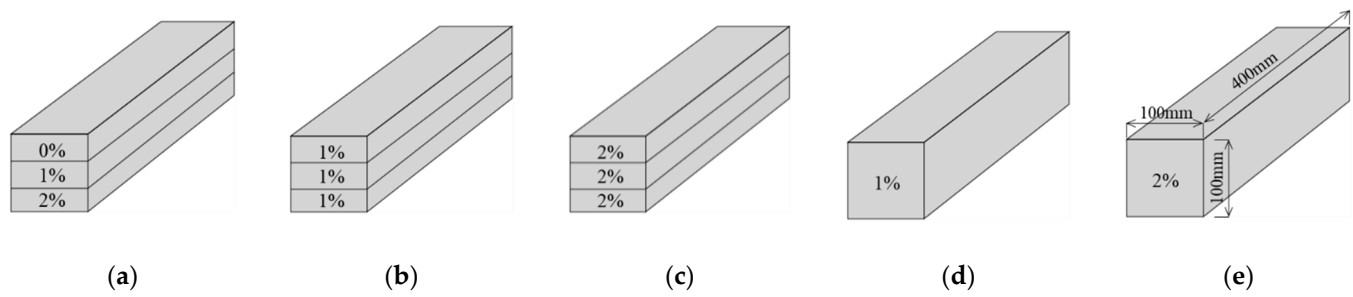

| (a) | (b) | (c) | (d) | (e) |

**Figure 1.** Specimens: (**a**) fiber-reinforced cementitious composite (FG-FRCC); (**b**) Layer-1%; (**c**) Layer-2%; (**d**) Hmg-1%; (**e**) Hmg-2%.

### 2.3. Specimen Fabrication

In the cases of FG-FRCC and layered specimens, there are interfacial transition zones among the layers. Shear stress is transmitted through the interfacial transition zones. In the case of the FG-FRCC specimen, furthermore, different mechanical properties in each layer are due to the use of different fiber volume fraction that causes large shear stress [18]. It has a possibility to fail by shear at the interfacial transition zones. The previous study [19], in which the extrusion fabrication method was adopted to make thin FRCC plates, introduced the "stacking and pressing" method to form the whole specimen from four plates. It was said that a strong inter-layer bond developed later through cement hydration.

In this study, the simplest fabrication method is conducted to be able to adopt in the practical use. Fresh FRCC is poured into the mold, making the most of the advantages of self-consolidating property of the FRCC used in this study. The poured FRCC causes self-leveling, so the layer is formed automatically. If the time intervals between the pouring for the next layer are too long, a cold joint would be generated between the layers. If the time intervals are too short, in contrast, pouring the next layer would disarrange the first layer FRCC. The time intervals between the layers pouring were set to 60 min considering the initial setting time of cementitious matrix. The bonds between layers are expected by cement hydration.

Figure 2 shows the procedures of the casting. The target height (33 mm thickness for each layer) was labeled inside the mold (Figure 2a). After pouring the first layer (Figure 2b), pouring for the second layer was carried out 60 min after the first layer pouring (Figure 2c). The temperature of fresh FRCC ranged from 12.0 to 14.1 degrees Celsius.

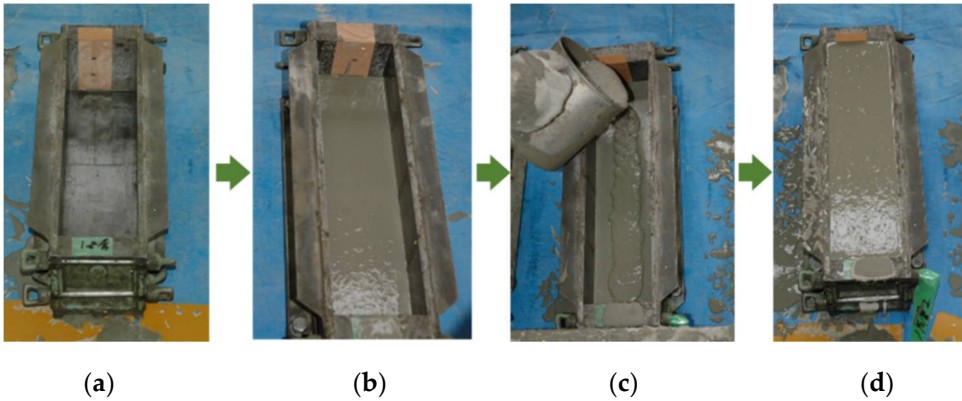

**Figure 2.** Fabrication of specimen: (**a**) mold before casting; (**b**) after pouring the first layer; (**c**) pouring the second layer 60 min after the first layer pouring; (**d**) after pouring the third layer.

## 2.4. Loading and Measurement

A four-point bending test is performed according to ISO 21914 [29]. Pure flexural characteristics and the average curvature in the constant bending moment region can be obtained by the four-point bending test. Though the loading surface is specified as being perpendicular to the casting surface in ISO 21914, the loading is carried out from the top side of the specimen. Figure 3 shows the positions of the linear variable displacement transducers (LVDTs) to measure deflections and curvature. Two π type LVDTs were fixed on the front surface of the specimen, and axial deformations in the constant bending moment region were measured. The average curvature can be obtained from the difference between the upper and lower strains by axial deformations. Three LVDTs were placed at the loading points and mid-span on the back surface to measure deflection of the specimen.

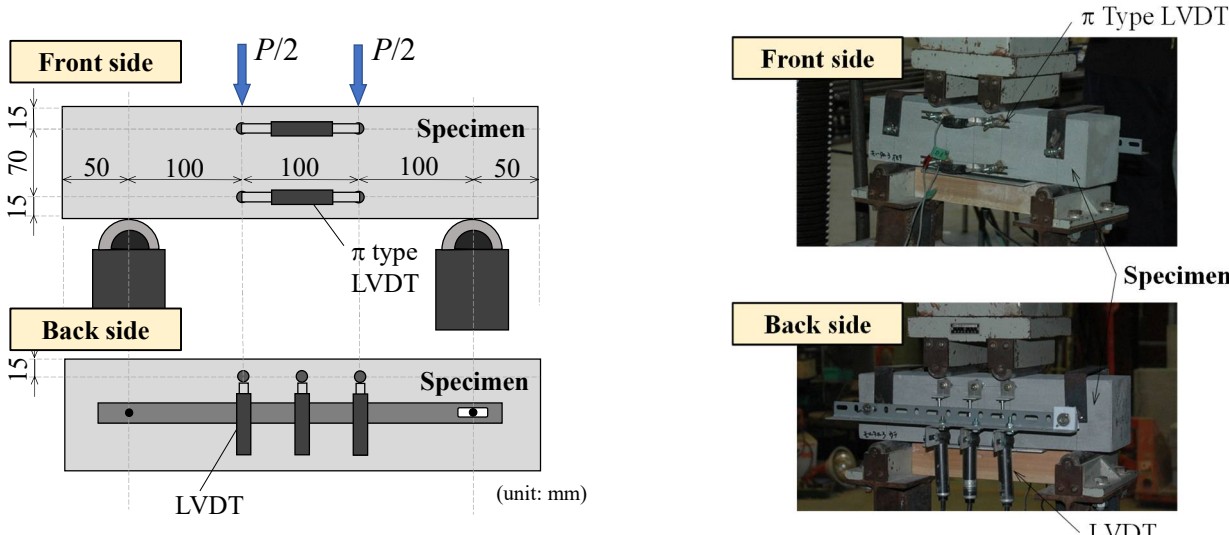

**Figure 3.** Four-point bending test setup and positions of linear variable displacement transducers (LVDTs).

Displacement controlled universal loading machine was utilized. The monotonic loading with the constant loading speed of 0.5 mm/min was carried out. The applied load and displacements by LVDTs were recorded by the data logger in every two seconds. The room temperature at the loading days ranged from 6.7 to 13.0 degrees Celsius.

## 3. Experimental Results

### 3.1. Failure Pattern

Figure 4 shows the typical examples of the specimens after loading. The black lines show cracks traced by the visible observation. All specimens exhibit ductile behavior by the bridging effect of fiber. The specimens with 1% volume fraction failed showing one or two cracks. The specimens with 2% volume fraction and FG-FRCC specimens presented three to five cracks. The crack opening was localized into one crack, and the load reached the maximum. Compressive failure on the compression edge could not be observed.

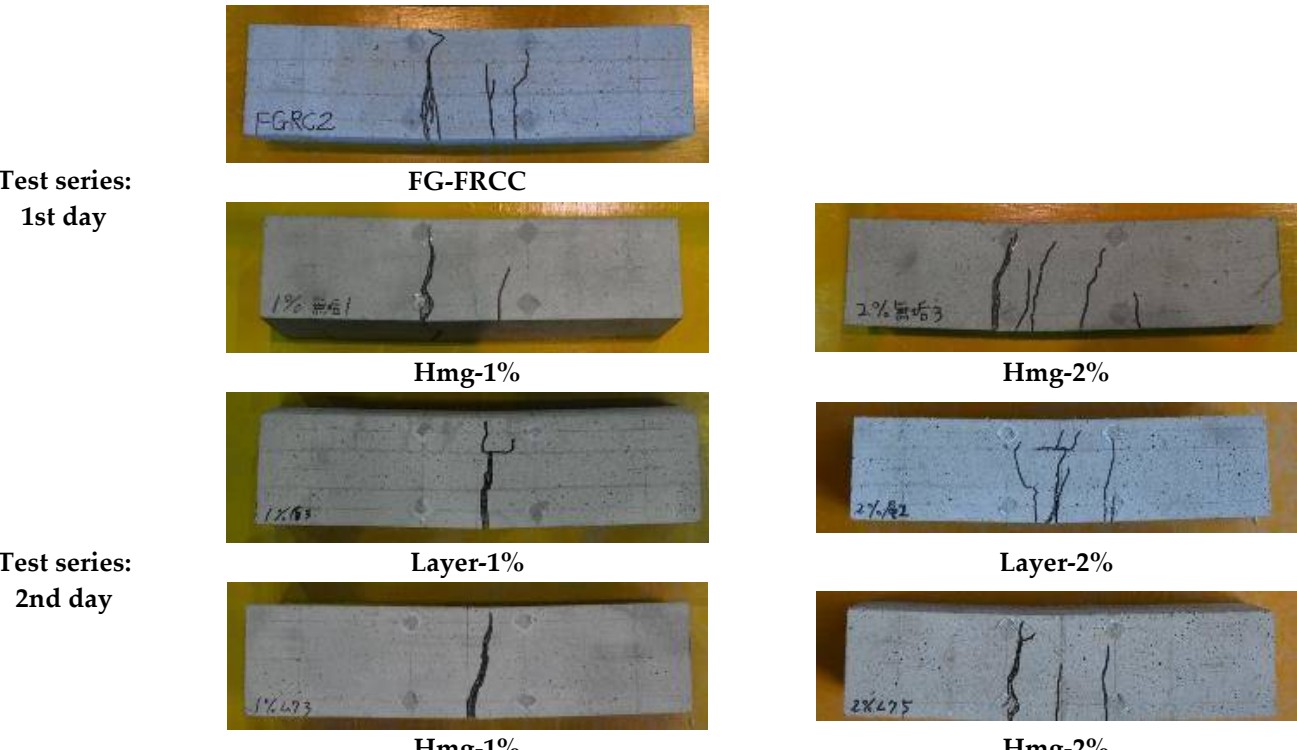

**Figure 4.** Typical examples of the specimens after loading.

There was no clear separation between the layers in the FG-FRCC and the layered specimens. Specimens Layer-1% and Layer-2% showed a small crack between the second and third layer, at which the first crack propagation stopped. It is considered that the bond between layers is enough to transmit shear stress under pure bending by the pouring fabrication method adopted in this study.

### 3.2. Load–Deflection Curve

Figures 5 and 6 show the applied load–deflection curves of the 1st day series and 2nd day series specimens, respectively. The deflection is the midspan deflection measured by the middle LVDT shown in Figure 3. The deflection-hardening behavior, that exhibits the load increasing after first cracking, can be recognized in all specimens. The steps in post-peak branch can be seen due to the propagation of existing cracks because of the displacement-controlled loading. Some scatterings of curves around the maximum load and post-peak branch among the same parameter specimens were observed in FG-FRCC and Hmg-2% specimens. Other series of specimens show little dispersion of curves among three specimens except 2nd day Layer-2% specimens. One of the 2nd day Layer-2% specimens failed by a localized crack opening out of the constant bending moment region.

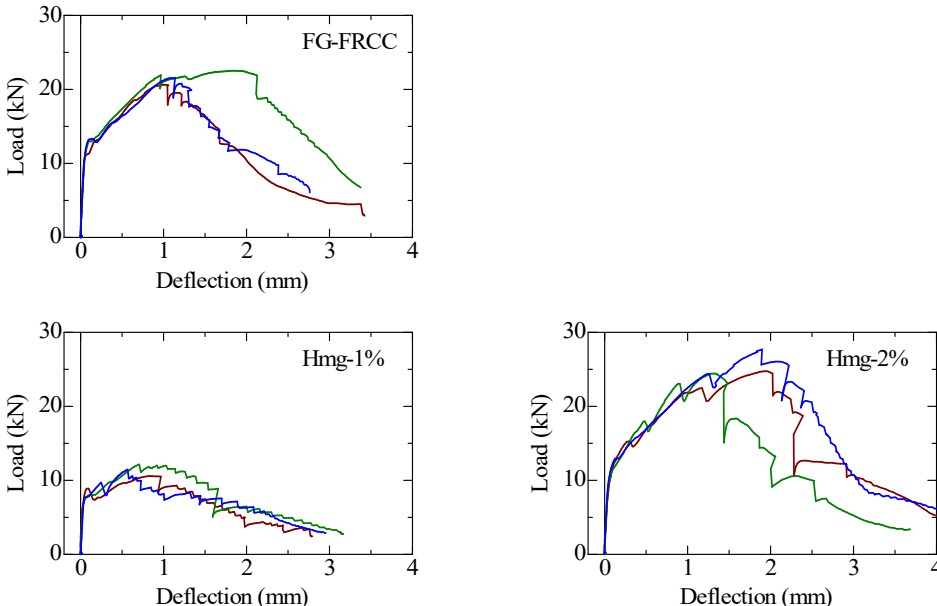

**Figure 5.** Applied load–deflection curves of 1st day series specimens.

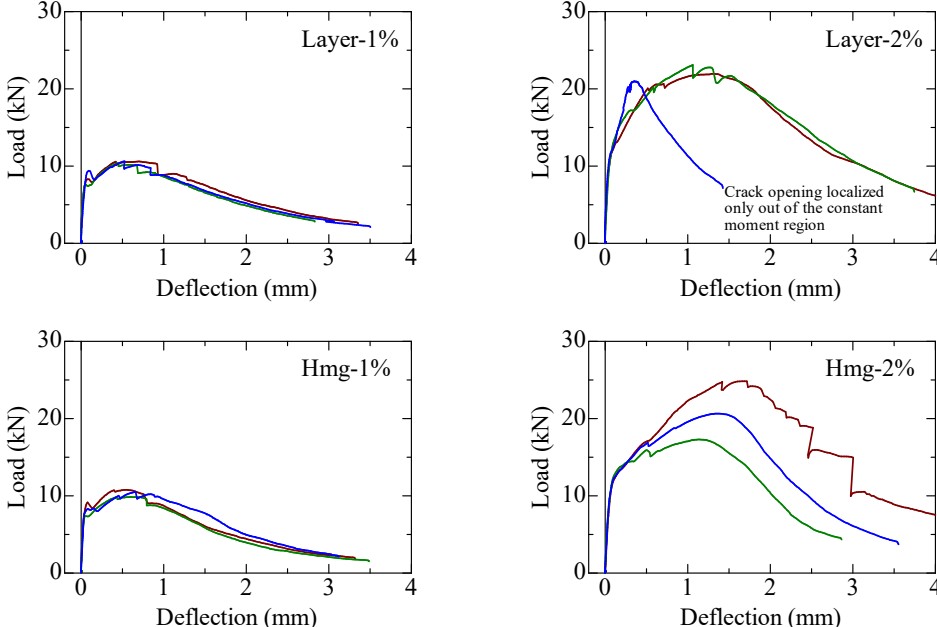

**Figure 6.** Applied load–deflection curves of 2nd day series specimens.

### 3.3. Maximum Load

Figure 7 shows the maximum load of all specimens. The plot for the Layer-2% specimen which failed by a localized crack opening out of the constant bending moment region is excluded. The values indicated at the top of bars are the averages of the maximum loads among the same parameters.

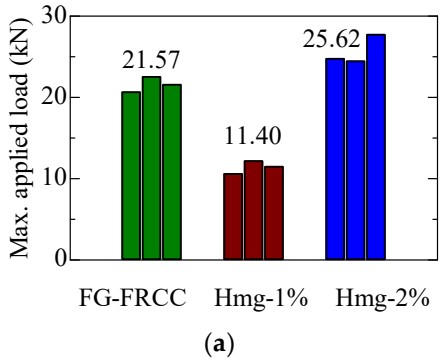
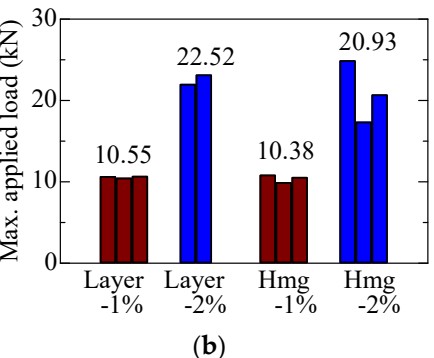

**Figure 7.** Maximum load comparison: (**a**) 1st day series specimen; (**b**) 2nd day series specimen.

The maximum load of the FG-FRCC specimens exhibits almost twice that of Hmg-1% specimens even the average of fiber volume fraction in whole specimen is the same as 1%. The maximum load of FG-FRCC specimens is smaller than that of Hmg-2% specimens. It is considered that the difference between the maximum loads of FG-FRCC and Hmg-2% specimens is due to the contribution of the fiber in the middle layer, in which the fiber volume fraction is 1% and 2% in FG-FRCC and Hmg-2%, respectively.

By the results of comparing the 2nd day series specimens, the effect of the layer is considered as not large. The ratio of the maximum load of the layered specimens to that of homogeneous ones is 1.02 and 1.08 for 1% and 2% volume fraction, respectively. The thickness of each layer adopted in this study is 33 mm in contrast to the fiber length of 12 mm. The thinner thickness may be required to show more effective contribution of the layer causing two-dimensional fiber orientation. However, increasing the number of the layers brings the complex fabrication of the specimen and requires much time.

## 4. Section Analysis and Comparison with Experimental Results

Section analysis is conducted to find better understandings of the effect of functionally graded layers in the flexural field. The stress–strain models based on the bridging law (tensile stress–crack width relationship owned by fibers) considering fiber orientation effect for PVA-FRCC are utilized in the analysis.

### 4.1. Method of Section Analysis

The section analysis is carried out for finite elements on the section of the specimen based on the assumption that the plain section remains plain. The stress–strain models for the tension side are derived from the bridging law for PVA-FRCC proposed by the authors [31]. The proposed tri-linear model for the bridging law is shown in Figure 8.

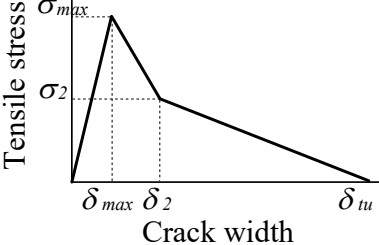

**Figure 8.** Tri-linear model for bridging law of PVA-FRCC [31].

The model has been led by the bridging law calculation [27] in which the pullout loads of individual fibers through crack surface in matrix are added considering the influence of fiber orientation. The characteristic points, $\sigma_{max}$, $\sigma_2$, and $\delta_{max}$ in tri-linear model are given

by following equations as functions of orientation intensity, $k$. These equations have been proposed for same PVA fiber used in this study, and for fiber volume fraction of 2%.

$$\sigma_{max} = 2.0 \, k^{0.30} \text{ (MPa)} \tag{1}$$

$$\sigma_2 = 0.60 \, k^{0.73} \text{ (MPa)} \tag{2}$$

$$\delta_{max} = 0.20 \, k^{0.18} \text{ (mm)} \tag{3}$$

The orientation intensity, $k$, presents the tendency of fiber orientation in matrix. The random orientation of fibers is given by $k = 1$. When the value of $k$ becomes larger than 1, fibers tend to orient toward the principal orientation angle (axial direction of specimen). When the value of $k$ becomes smaller than 1, fibers tend to orient toward the perpendicular to the principal orientation angle. The detailed explanations about orientation intensity and principal orientation angle can be found in the literature [27]. The value for $\delta_2$ has been proposed as 0.45 mm, that corresponds to the crack width at which the individual fiber shows the maximum pullout load [31]. The reason that the maximum tensile stress in the bridging law reaches before $\delta_2$, is that individual fibers start rupturing before showing the maximum pullout load due to the reduction of the apparent rupture strength by orientation angle in the case of polymeric fibers [32]. The crack width $\delta_{tu}$, at which the tensile stress vanishes, is given as the half length of fiber. All fibers are completely pulled out when the crack width reaches the half of fiber length.

The stress–strain models for section analysis are shown in Figure 9. The parabolic curves are selected for the compressive stress–strain model. The compressive strengths and strains at the maximum are based on the compression test results. The tensile stress–strain models are given by the bridging laws calculated as described before. The orientation intensity is assumed to be same value of 6.5 in the previous study [27], that is the averaged value obtained by the side views in visualization simulation for horizontal casting similar with this study. The strains are converted from crack width divided by the length of constant bending moment region, that is equal to the gauge length of LVDTs. The tensile stress for 1% fiber volume fraction is set to half that for 2% volume fraction. In the case of 0% volume fraction (mortar), the tensile strength is assumed to 2 MPa at 0.011% strain in accordance with the elastic modulus measured by compression test. After the tensile strength, linear softening branch, that the stress becomes to zero at 0.022% strain, is adopted.

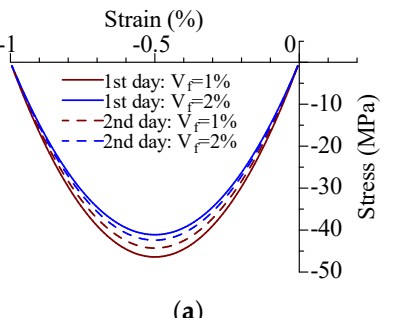

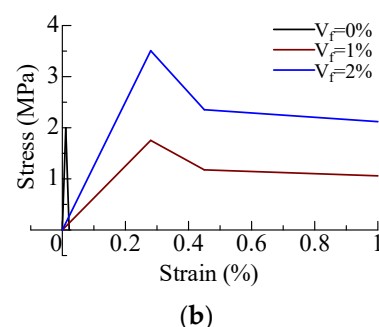

(**a**)          (**b**)

**Figure 9.** Stress-strain model for section analysis: (**a**) compression side; (**b**) tension side.

### 4.2. Comparison of Maximum Bending Moment

Table 4 list the analysis results with the experimental results of maximum bending moment. The experimental results are obtained by following equation.

$$_e M_{max} = P_{max} \cdot l / 6 \tag{4}$$

where, $_e M_{max}$: experimental maximum bending moment, $P_{max}$: maximum load, $l$: span length.

**Table 4.** List of maximum bending moment.

| Test Series (Casting Day) | Specimen ID | Comp. Strength (MPa) | Experiment Max. Bending Moment $_eM_{max}$ (kN·m) Avg. | STDV | Curvature at $_eM_{max}$ (μ/mm) | Section Analysis Max. Bending Moment $_aM_{max}$ (kN·m) | Neutral Axis from Comp. Edge (mm) | Ratio of Experiment to Analysis $_eM_{max}/_aM_{max}$ |
|---|---|---|---|---|---|---|---|---|
| 1stday | FG-FRCC | [1] | 1.079 | 0.047 | 209 | 0.950 | 19.5 | 1.14 |
|  | Hmg-1% | 46.4 | 0.570 | 0.079 | 129 | 0.583 | 14.7 | 0.98 |
|  | Hmg-2% | 41.1 | 1.281 | 0.090 | 178 | 1.087 | 20.6 | 1.18 |
| 2ndday | Layer-1% | 44.3 | 0.528 | 0.006 | 99 | 0.581 | 15.0 | 0.91 |
|  | Layer-2% | 42.4 | 1.126 | 0.041 | 157 | 1.090 | 20.3 | 1.03 |
|  | Hmg-1% | 44.3 | 0.519 | 0.024 | 107 | 0.581 | 15.0 | 0.89 |
|  | Hmg-2% | 42.4 | 1.047 | 0.189 | 205 | 1.090 | 20.3 | 0.96 |

[1] Compressive strengths for $V_f$ = 0%, 1%, and 2% are 40.9, 46.4, and 41.1 MPa, respectively.

The ratio of the experimental maximum bending moment to the analysis result ranges from 0.89 to 1.18. It is considered that the section analysis result considering the fiber orientation in the bridging laws show a good adaptability with the experiment result. The analysis result also indicates that the maximum moment of FG-FRCC specimen is 1.63 times that of the homogeneous specimen with the same whole fiber volume fraction of 1%. Though the maximum moments of layered specimens in the experiment are slightly larger than for the homogeneous specimens, the analytical results have no difference between the layered and homogeneous specimens because of using the same models. If the fiber orientation intensity in the layered specimen can be evaluated, the difference in analysis has a possibility to be presented by the bridging laws with the appropriate fiber orientation, e.g., considering two or three-dimensional orientation and the wall effect [23].

Figure 10 shows stress distributions in cross-sections of the specimens at showing the maximum bending moment. The horizontal axis indicates the stress of FRCC, showing compression in the left (negative) and tension in the right (positive). In the homogeneous specimens, the maximum bending moment is demonstrated when the maximum tensile stress positions at around one-third of the whole height of the cross-section from the tension edge. In contrast, the tensile stress shows the maximum at about half the height in the 1st layer (2% volume fraction) of FG-FRCC specimen. It is considered that the bending moment reaches the maximum when the tensile force in the tension side layer becomes maximum in the case of FG-FRCC.

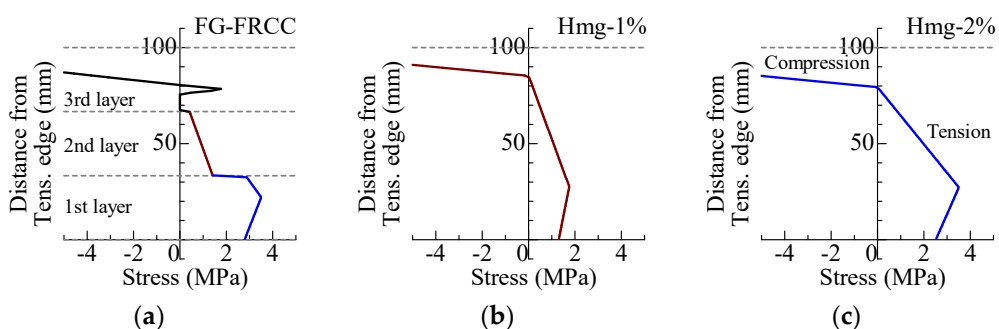

**Figure 10.** Stress distribution in cross-section at the maximum moment: (**a**) 1st day FG-FRCC; (**b**) 1st day Hmg-1%; (**c**) 1st day Hmg-2%.

## 5. Conclusions

Based on the results of the experiment and section analysis conducted to investigate the flexural characteristics of the FG-FRCC, the following conclusions are drawn:

1.  Clear separation between layers in the FG-FRCC was not observed in the four-point bending test. It is considered that the bond between layers is enough to transmit shear

stress under pure bending by the pouring fabrication method for PVA-FRCC having self-consolidating properties.

2.  The maximum load of the FG-FRCC specimens exhibited almost twice that of the homogeneous specimens, even when the average of fiber volume fraction in whole specimen is 1%.

3.  The ratio of the maximum load of the three-layered specimens with the same fiber volume fraction to that of the homogeneous specimens is 1.02 and 1.08 for the 1% and 2% volume fractions, respectively. The thinner thickness may be required to show the more effective contribution of the layer causing the two-dimensional fiber orientation.

4.  Section analysis, in which the stress–strain models based on the bridging law considering the fiber orientation effect was conducted. The ratio of the experimental maximum moment to the analysis result ranges from 0.89 to 1.18. It is considered that the section analysis result considering the fiber orientation shows a good adaptability with the experiment result.

5.  The analysis result shows that the maximum moment of the FG-FRCC specimen is 1.63 times that of the homogeneous specimen with the same whole fiber volume fraction of 1%. It is considered that the bending moment reaches the maximum when the tensile force in the tension side layer becomes maximum in the case of the FG-FRCC.

FG-FRCC has a possibility to use fibers as one of the cost-effective FRCCs that exhibits higher strength compared to homogeneous FRCC with the same overall fiber volume fraction. This study also indicates the simplest casting method in which self-consolidated FRCC is poured into the mold for each layer with appropriate time intervals. It is considered that this method can be easily adopted in the practical use of FG-FRCC.

**Author Contributions:** Conceptualization, T.K. (Toshiyuki Kanakubo) and T.K. (Takumi Koba); methodology, T.K. (Toshiyuki Kanakubo) and T.K. (Takumi Koba); validation, T.K. (Toshiyuki Kanakubo); formal analysis, T.K. (Takumi Koba) and K.Y.; investigation, T.K. (Takumi Koba) and K.Y.; writing—original draft preparation, T.K. (Toshiyuki Kanakubo); writing—review and editing, T.K. (Takumi Koba) and K.Y.; visualization, T.K. (Toshiyuki Kanakubo); supervision, T.K. (Toshiyuki Kanakubo); project administration, T.K. (Toshiyuki Kanakubo); funding acquisition, T.K. (Toshiyuki Kanakubo). All authors have read and agreed to the published version of the manuscript.

**Funding:** This research was funded by the JSPS KAKENHI Grant Number 18H03802.

**Data Availability Statement:** Data available on request.

**Conflicts of Interest:** The authors declare no conflict of interest. The funders had no role in the design of the study; in the collection, analyses, or interpretation of data; in the writing of the manuscript, or in the decision to publish the results.

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
