# Peer review of "Flexural Characteristics of Functionally Graded Fiber-Reinforced Cementitious Composite with Polyvinyl Alcohol Fiber"

_jcs, doi:10.3390/jcs5040094_

Round 1
Reviewer 1 Report
Overall, the research work is well conducted, I got few suggestions and hope the authors can consider:
In the engineering materials fields, FGMs have been developed for the industrial applications such as aerospace, automotive, power generation, and so on. Could the authors kindly add few references for these applications? e.g. for aerospace, it would be helpful to add: 10.1016/j.compstruct.2015.08.113; 10.1016/j.compositesb.2015.07.018.
Please point out the research gap before stating the objective/aim of this work. After all the literature review, what is the reason for doing this research? What is the main novelty compared to Ref [17]?
Fresh FRCC is poured into the mold… What is the temperature of the fresh FRCC before pouring? Is it controlled for each pouring? What is the layer thickness and how is it controlled?
What is the temperature for four-point bending test?
Some labels could be added to the practical experiment shown in Fig. 3. Also some figure captions are needed. What is the different between these two photographs?
Reviewer 2 Report
The authors in their work focused on presenting the results concerning investigate the flexural characteristics of functionally graded fiber-reinforced cementitious composite (FG-FRCC) in the viewpoints of fiber volume fraction (Vf) varying in layers and layered effect in bending specimens. In this experiment, all specimens were fabricated by pouring FRCC into the molds. I think that the studies are interesting. In my opinion, this manuscript does provide sufficient important knowledge in composites science to be considered for publication. The topics discussed in this manuscript fully match the magazine. In my judgment that the authors in an interesting way to present a study to described the flexural characteristics of FG-FRCC. Clear separation between layers in FG-FRCC was not observed in the four-point bending test. This may indicate that the bond between layers is enough to transmit shear stress under pure bending by the pouring fabrication method for PVA-FRCC having self-consolidating property. The authors have conducted section analysis, in which they used the stress-strain models based on the bridging law considering fiber orientation effect. On this analysis, they concluded that the section analysis effect taking into account fiber orientation shows good adaptability with the experiment result. I want to press the point that the presented data are reliable and useful for other researchers in the world. The title of the manuscript is acceptable - it fully describes the issues presented in the article. From my point of view, the authors of the manuscript described the most covers pertinent points in section "abstract". In my opinion, the authors were designed and conducted the research included in the article in detail. The authors have described in detail the method of preparation of samples and taking measurements. There is no doubt that they presented the obtained results reliably and clearly. In my opinion, the scientific quality of the manuscript is good. The figures & tables included in the manuscript are of good quality and require no corrections. I conclude that conclusions are adequately supported by the data. References are adequately cited by the authors. The level of auto-citation is equal to 13%. Unfortunately, the language needed a little correction. In my opinion, the paper is accepted with minor corrections.
Author Response
Thank you so much for your positive comments. The language has been reviewed again carefully including the native's check, and improved.
Reviewer 3 Report
Comments
This paper studied the flexural characteristics of FRP beams. The outcome is interesting for readers. However, there are several aspects that need to be improved. The reviewer can only recommend for publication if the author satisfactorily address the following comments in the revised version.
- The shear span-to-depth ratio is very small for flexural test. Why the authors selected 300 mm span for flexural test?
- Why 4-point bending test was selected over 3-point bending? Need a justification.
- How many specimens were tested for each case? Suggest to provide standard deviation if it is more than one.
- 7, the maximum load in Y-axis should be replaced by bending strength. Strength is more realistic than load.
- What causes non-linearity in load-deflection curve in Fig. 5 and Fig. 6?
- The failure mechanism of the specimen should be discussed more clearly.
- The novelty of the study should be highlighted at the end of introduction section. How this study is different from the published study in literature?
- How the outcome of this study will benefit researchers and end users? This need to be highlighted in introduction or end of conclusion.
- The background study on the application of FRP cementitious composites is insufficient. Recently, FRP cementitious composites was used for manufacturing railway sleepers [Ref: Static behaviour of glass fibre reinforced novel composite sleepers for mainline railway track], column jackets [Ref: State-of-the-art of prefabricated FRP composite jackets for structural repair], and also applied in 3D printed concrete [Ref: 3D-printed concrete: applications, performance, and challenges]. Suggest to include them in introduction section with proper citations to improve the background study.
I would be happy to see the revised version to understand how these comments are being addressed.
Round 2
Reviewer 1 Report
Thank you for the response, I have no further comments.
Reviewer 3 Report
Some comments such as novelty of this study (point 7) and the use of short fibres in 3D printed concrete (point 9) are not convincingly addressed. Suggest to address them.